# Attention-over-Attention Neural Networks for Reading Comprehension

## Abstract

Cloze-style reading comprehension is a representative problem in mining relationship between document and query. In this paper, we present a simple but novel model called *attention-over-attention* reader for better solving cloze-style reading comprehension task. Our model aims to place another attention mechanism over the document-level attention and induces "attended attention" for final answer predictions. One advantage of our model is that it is simpler than related works while giving excellent performance. We also propose an N-best re-ranking strategy to double check the validity of the candidates and further improve the performance. Experimental results show that the proposed methods significantly outperform various state-of-the-art systems by a large margin in public datasets, such as CNN and Children's Book Test.

## 1 Introduction

To read and comprehend the human languages are challenging tasks for the machines, which requires that the understanding of natural languages and the ability to do reasoning over various clues. Reading comprehension is a general problem in the real world, which aims to read and comprehend a given article or context, and answer the questions based on it. Recently, the cloze-style reading comprehension problem has become a popular task in the community. The cloze-style query (Taylor, 1953) is a problem that to fill in an appropriate word in the given sentences while taking the context information into account.

To teach the machine to do cloze-style reading comprehensions, large-scale training data is necessary for learning relationships between the given document and query. To create large-scale training data for neural networks, Hermann et al. (2015) released the CNN/Daily Mail news dataset, where the document is formed by the news articles and the queries are extracted from the summary of the news. Hill et al. (2015) released the Children's Book Test dataset afterwards, where the training samples are generated from consecutive 20 sentences from books, and the query is formed by 21st sentence. Following these datasets, a vast variety of neural network approaches have been proposed (Kadlec et al., 2016; Cui et al., 2016; Chen et al., 2016; Dhingra et al., 2016; Sordoni et al., 2016; Trischler et al., 2016), and most of them stem from the attention-based neural network (Bahdanau et al., 2014), which has become a stereotype in most of the NLP tasks and is well-known by its capability of learning the "importance" distribution over the inputs.

In this paper, we present a novel neural network architecture, called *attention-over-attention* model. As we can understand the meaning literally, our model aims to place another attention mechanism over the existing document-level attention. Unlike the previous works, that are using heuristic merging functions (Cui et al., 2016), or setting various pre-defined non-trainable terms (Trischler et al., 2016), our model could automatically generate an "attended attention" over various document-level attentions, and make a mutual look not only from *query-to-document* but also *document-to-query*, which will benefit from the interactive information.

To sum up, the main contributions of our work are listed as follows.

- To our knowledge, this is the first time that the mechanism of nesting another attention over the existing attentions is proposed, i.e.

*attention-over-attention* mechanism.

- Unlike the previous works on introducing complex architectures or many non-trainable hyper-parameters to the model, our model is much more simple but outperforms various state-of-the-art systems by a large margin.

- We also propose an N-best re-ranking strategy to re-score the candidates in various aspects and further improve the performance.

## 2 Cloze-style Reading Comprehension

In this section, we will give a brief introduction to the cloze-style reading comprehension task at the beginning. And then, several existing public datasets will be described in detail.

### 2.1 Task Description

Formally, a general Cloze-style reading comprehension problem can be illustrated as a triple:

$$\langle \mathcal{D}, \mathcal{Q}, \mathcal{A} \rangle$$

The triple consists of a document $\mathcal{D}$, a query $\mathcal{Q}$ and the answer to the query $\mathcal{A}$. Note that the answer is usually a *single* word in the document, which requires the human to exploit context information in both document and query. The type of the answer word varies from predicting a preposition given a fixed collocation to identifying a named entity from a factual illustration.

### 2.2 Existing Public Datasets

Large-scale training data is essential for training neural networks. Several public datasets for the cloze-style reading comprehension has been released. Here, we introduce two representative and widely-used datasets.

**CNN/Daily Mail.**[1] Hermann et al. (2015) have firstly published two datasets: CNN and Daily Mail news data. They construct these datasets with web-crawled CNN and Daily Mail news data. One of the characteristics of these datasets is that the news article is often associated with a summary. So they first regard the main body of the news article as the *Document*, and the *Query* is formed by the summary of the article, where one entity word is replaced by a special placeholder to indicate the missing word. The replaced entity word

will be the *Answer* of the *Query*. Apart from releasing the dataset, they also proposed a methodology that anonymizes the named entity tokens in the data, and these tokens are also re-shuffle in each sample. The motivation is that the news articles are containing limited named entities, which are usually celebrities, and the world knowledge can be learned from the dataset. So this methodology aims to exploit general relationships between anonymized named entities within a single document rather than the common knowledge. The following research on these datasets showed that the entity word anonymization is not that effective than expected (Chen et al., 2016).

**Children's Book Test.** [2] There was also a dataset called the Children's Book Test (CBTest) released by Hill et al. (2015), which is built on the children's book story through Project Gutenberg. Different from the CNN/Daily Mail datasets, there is no summary available in the children's book. So they proposed another way to extract query from the original data. The document is composed of 20 consecutive sentences in the story, and the 21st sentence is regarded as the query, where one word is blanked with a special placeholder. In the CBTest datasets, there are four types of sub-datasets available which are classified by the part-of-speech tag of the answer word, containing Named Entities (NE), Common Nouns (CN), Verbs and Prepositions. In their studies, they have found that the answering of verbs and prepositions are relatively less dependent on the content of document, and the humans can even do preposition blank-filling without the presence of the document. As the aim of reading comprehension is to exploit relations between document and query, most of the following studies are only focusing on the NE and CN datasets.

## 3 Attention-over-Attention Reader

In this section, we will give a detailed introduction to the proposed Attention-over-Attention Reader (AoA Reader). Our model is primarily motivated by Kadlec et al., (2016), which aims to directly estimate the answer from the document-level attention instead of calculating blended representations of the document. As previous studies by Cui et al. (2016) showed that the further investigation

---

[1]The pre-processed CNN and Daily Mail datasets are available at http://cs.nyu.edu/~kcho/DMQA/

[2]The CBTest datasets are available at http://www.thespermwhale.com/jaseweston/babi/CBTest.tgz

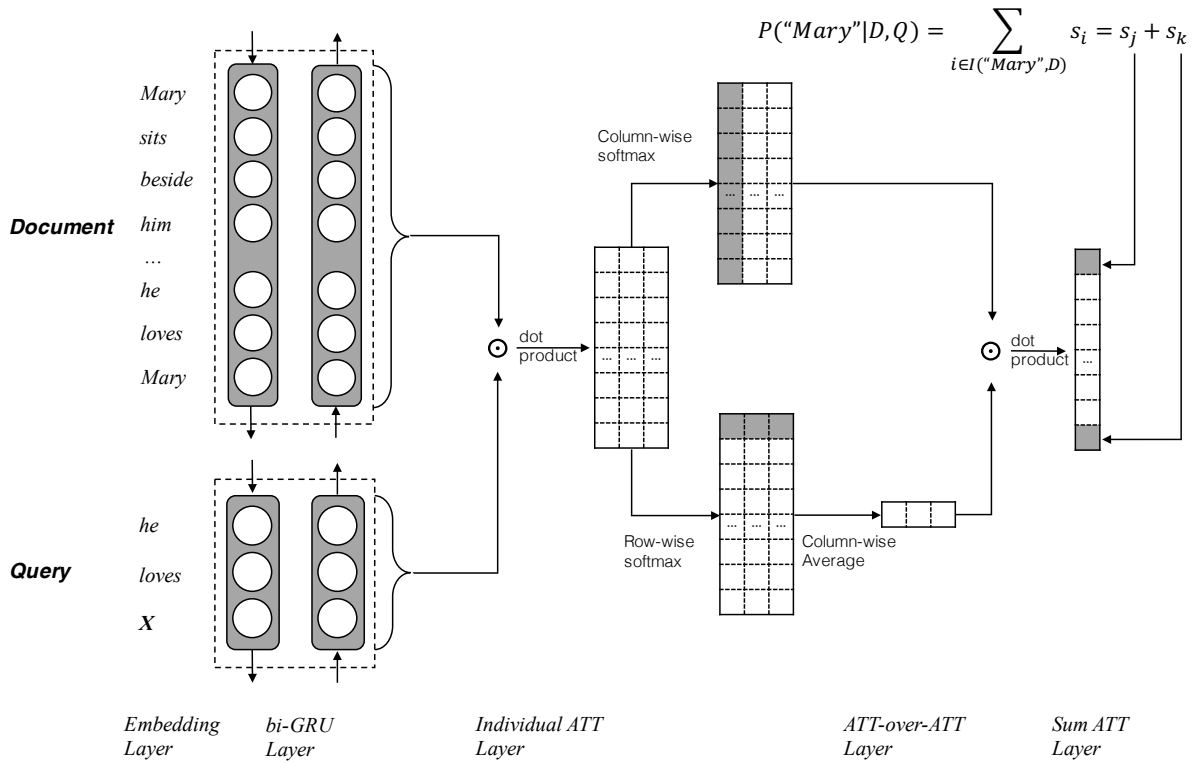

Figure 1: Neural network architecture of the proposed Attention-over-Attention Reader (AoA Reader).

of query representation is necessary, and it should be paid more attention to utilizing the information of query. In this paper, we propose a novel work that placing another attention over the primary attentions, to indicate the "importance" of each attentions.

Now, we will give a formal description of our proposed model. When a cloze-style training triple $\langle \mathcal{D}, \mathcal{Q}, \mathcal{A} \rangle$ is given, the proposed model will be constructed in the following steps.

**Contextual Embedding.** We first transform every word in the document $\mathcal{D}$ and query $\mathcal{Q}$ into one-hot representations and then convert them into continuous representations with a shared embedding matrix $W_e$. The motivation of using shared embedding weights is that the length of the query is shorter than the document, and thus the embedding weights will not be fully learned by only using a small amount of training data. By embedding sharing, both the document and query can participate in the learning of embedding and both of them will benefit from this mechanism. After that, we use two bi-directional RNNs to get contextual representations of the document and query individually, where the representation of each word

is formed by concatenating the forward and backward hidden states. After making a trade-off between model performance and training complexity, we choose the Gated Recurrent Unit (GRU) (Cho et al., 2014) as recurrent unit implementation.

$$e(x) = W_e \cdot x, \; where \; x \in \mathcal{D}, \mathcal{Q} \qquad (1)$$

$$\overrightarrow{h_s(x)} = \overrightarrow{GRU}(e(x)) \qquad (2)$$

$$\overleftarrow{h_s(x)} = \overleftarrow{GRU}(e(x)) \qquad (3)$$

$$h_s(x) = [\overrightarrow{h_s(x)}; \overleftarrow{h_s(x)}] \qquad (4)$$

We take $h_{doc} \in \mathbb{R}^{|\mathcal{D}|*2d}$ and $h_{query} \in \mathbb{R}^{|\mathcal{Q}|*2d}$ to denote the contextual representations of document and query, where $d$ is the dimension of GRU (one-way).

**Pair-wise Matching Score.** After obtaining the contextual embeddings of the document $h_{doc}$ and query $h_{query}$, we calculate a pair-wise matching matrix, which indicates the pair-wise matching degree of one document word and one query word. Formally, when given $i$th word of the document and $j$th word of query, we can compute a matching score by their dot product.

$$M(i,j) = h_{doc}(i)^T \cdot h_{query}(j) \qquad (5)$$

In this way, we can calculate every pair-wise matching score between each document and query word, forming a matrix $M \in \mathbb{R}^{|\mathcal{D}|*|\mathcal{Q}|}$, where the value of $i$th row and $j$th column is filled by $M(i, j)$.

**Individual Attentions.** After getting the pair-wise matching matrix $M$, we apply a column-wise soft-max function to get probability distributions in each column, where each column is an individual document-level attention when considering a single query word. We denote $\alpha(t) \in \mathbb{R}^{|\mathcal{D}|}$ as the document-level attention regarding query word at time $t$, which can be seen as a *query-to-document* attention.

$$\alpha(t) = softmax(M(1, t), ..., M(|\mathcal{D}|, t)) \quad (6)$$
$$\alpha = [\alpha(1), \alpha(2), ..., \alpha(|\mathcal{Q}|)] \quad (7)$$

**Attention-over-Attention.** Instead of using naive heuristics (such as *summing* or *averaging*) to combine these individual attentions into a final attention, we introduce another attention mechanism to automatically decide the importance of each individual attention.

First, we calculate a reversed attention, that is, for every document word at time $t$, we calculate the "importance" distribution on the query, to indicate which query words are more important given a single document word. We apply a row-wise softmax function to the pair-wise matching matrix $M$ to get query-level attentions. We denote $\beta(t) \in \mathbb{R}^{|\mathcal{Q}|}$ as the query-level attention regarding document word at time $t$, which can be seen as a *document-to-query* attention.

$$\beta(t) = softmax(M(t, 1), ..., M(t, |\mathcal{Q}|)) \quad (8)$$

So far, we have obtained both *query-to-document* attention $\alpha$ and *document-to-query* attention $\beta$. Our motivation is to exploit mutual information between the document and query. However, most of the previous works are only relying on *query-to-document* attention, that is, only calculate one document-level attention when considering the whole query.

Then we average all the $\beta(t)$ to get an averaged query-level attention $\beta$. Note that, we do not apply another softmax to the $\beta$, because averaging individual attentions do not break the normalizing condition.

$$\beta = \frac{1}{n} \sum_{t=1}^{|\mathcal{D}|} \beta(t) \quad (9)$$

Finally, we calculate dot product of $\alpha$ and $\beta$ to get the "attended document-level attention" $s \in \mathbb{R}^{|\mathcal{D}|}$, i.e. the *attention-over-attention* mechanism. Intuitively, this operation is calculating a weighted sum of each individual document-level attention $\alpha(t)$ when looking at query word at time $t$. In this way, the contributions by each query word can be learned explicitly, and the final decision (document-level attention) is made through the voted result by the importance of each query word.

$$s = \alpha^T \beta \quad (10)$$

**Final Predictions.** Following Kadlec et al. (2016), we use *sum attention* mechanism to get aggregated results. Note that the final output should be reflected in the vocabulary space $V$, rather than document-level attention $|\mathcal{D}|$, which will make a significant difference in the performance, though Kadlec et al. (2016) did not illustrate this clearly.

$$P(w|\mathcal{D}, \mathcal{Q}) = \sum_{i \in I(w, \mathcal{D})} s_i, \ w \in V \quad (11)$$

where $I(w, \mathcal{D})$ indicate the positions that word $w$ appear in the document $\mathcal{D}$. As the training objectives, we seek to maximize the log-likelihood of the correct answer.

$$\mathcal{L} = \sum_i \log(p(x)) \ , x \in \mathcal{A} \quad (12)$$

The proposed neural network architecture is depicted in Figure 1. Note that, as our model mainly adds limited steps of calculations to the AS Reader (Kadlec et al., 2016) and do not employ any additional weights, the computational complexity is similar to the AS Reader.

## 4 N-best Re-ranking Strategy

Intuitively, when we do cloze-style reading comprehensions, we often refill the candidate into the blank of the query to double-check its appropriateness, fluency and grammar to see if the candidate we choose is the most suitable one. If we do find some problems in the candidate we choose, we will choose the second possible candidate and do some checking again.

To mimic the process of double-checking, we propose to use N-best re-ranking strategy after generating answers from our neural networks. The procedure can be illustrated as follows.

**N-best decoding.** Instead of only picking the candidate that has the highest possibility as answer,

| | CNN News | | | CBT NE | | | CBT CN | | |
|---|---|---|---|---|---|---|---|---|---|
| | Train | Valid | Test | Train | Valid | Test | Train | Valid | Test |
| # Query | 380,298 | 3,924 | 3,198 | 108,719 | 2,000 | 2,500 | 120,769 | 2,000 | 2,500 |
| Max # candidates | 527 | 187 | 396 | 10 | 10 | 10 | 10 | 10 | 10 |
| Avg # candidates | 26 | 26 | 25 | 10 | 10 | 10 | 10 | 10 | 10 |
| Avg # tokens | 762 | 763 | 716 | 433 | 412 | 424 | 470 | 448 | 461 |
| Vocabulary | | 118,497 | | | 53,063 | | | 53,185 | |

Table 1: Statistics of cloze-style reading comprehension datasets: CNN news and CBTest NE / CN.

we can also extract follow-up candidates in the decoding process, which forms an N-best list.

**Refill the candidate into query.** As a characteristic of the cloze-style problem, each candidate can be refilled into the blank of the query to form a complete sentence. This allows us to check the candidate according to its context.

**Feature scoring.** The candidate sentences can be scored in many aspects. In this paper, we exploit three features to score the N-best list.

- Global N-gram LM: This is a fundamental metric in scoring sentence, which aims to evaluate its fluency. This model is trained on the document part of training data.

- Local N-gram LM: Different from global LM, the local LM aims to explore the information with the given document, so the statistics are obtained from the test-time document. It should be noted that the local LM is trained sample-by-sample, it is not trained on the entire test set, which is not legal in the real test case. This model is useful when there are many unknown words in the test sample.

- Word-class LM: Similar to global LM, the word-class LM is also trained on the document part of training data, but the words are converted to its word class ID. The word class can be obtained by using clustering methods. In this paper, we simply utilized the *mkcls* tool for generating 1000 word classes (Josef Och, 1999).

**Weight Tuning.** To tune the weights among these features, we adopt the K-best MIRA algorithm (Cherry and Foster, 2012) to automatically optimize the weights on the validation set, which is widely used in statistical machine translation tuning procedure.

**Re-scoring and Re-ranking.** After getting the weights of each feature, we calculate the weighted sum of each feature in the N-best sentences and then choose the candidate that has the lowest cost as the final answer.

## 5 Experiments

### 5.1 Experimental Setups

The general settings of our neural network model are listed below in detail.

- Embedding Layer: The embedding weights are randomly initialized with the uniformed distribution in the interval $[-0.05, 0.05]$. For regularization purpose, we adopted $l_2$-regularization to 0.0001 and dropout rate of 0.1 (Srivastava et al., 2014). Also, it should be noted that we do not exploit any pre-trained embedding models.

- Hidden Layer: Internal weights of GRUs are initialized with random orthogonal matrices (Saxe et al., 2013).

- Optimization: We adopted ADAM optimizer for weight updating (Kingma and Ba, 2014), with an initial learning rate of 0.001. As the GRU units still suffer from the gradient exploding issues, we set the gradient clipping threshold to 5 (Pascanu et al., 2013). We used batched training strategy of 32 samples.

| | Embed. # units | Hidden # units |
|---|---|---|
| CNN News | 384 | 256 |
| CBTest NE | 384 | 384 |
| CBTest CN | 384 | 256 |

Table 2: Embedding and hidden layer dimensions for each task.

| | CNN News | | CBTest NE | | CBTest CN | |
|---|---|---|---|---|---|---|
| | Valid | Test | Valid | Test | Valid | Test |
| Deep LSTM Reader (Hermann et al., 2015) | 55.0 | 57.0 | - | - | - | - |
| Attentive Reader (Hermann et al., 2015) | 61.6 | 63.0 | - | - | - | - |
| Human (context+query) (Hill et al., 2015) | - | - | - | *81.6* | - | *81.6* |
| MemNN (window + self-sup.) (Hill et al., 2015) | 63.4 | 66.8 | 70.4 | 66.6 | 64.2 | 63.0 |
| AS Reader (Kadlec et al., 2016) | 68.6 | 69.5 | 73.8 | 68.6 | 68.8 | 63.4 |
| CAS Reader (Cui et al., 2016) | 68.2 | 70.0 | 74.2 | 69.2 | 68.2 | 65.7 |
| Stanford AR (Chen et al., 2016) | 72.4 | 72.4 | - | - | - | - |
| GA Reader (Dhingra et al., 2016) | 73.0 | 73.8 | 74.9 | 69.0 | 69.0 | 63.9 |
| Iterative Attention (Sordoni et al., 2016) | 72.6 | 73.3 | 75.2 | 68.6 | 72.1 | 69.2 |
| EpiReader (Trischler et al., 2016) | **73.4** | 74.0 | 75.3 | 69.7 | 71.5 | 67.4 |
| **AoA Reader** | 73.1 | **74.4** | **77.8** | 72.0 | 72.2 | **69.4** |
| **AoA Reader + Reranking** | - | - | 79.6 | 74.0 | 75.7 | 73.1 |
| MemNN (Ensemble) | 66.2 | 69.4 | - | - | - | - |
| AS Reader (Ensemble) | 73.9 | 75.4 | 74.5 | 70.6 | 71.1 | 68.9 |
| GA Reader (Ensemble) | 76.4 | 77.4 | 75.5 | 71.9 | 72.1 | 69.4 |
| EpiReader (Ensemble) | - | - | 76.6 | 71.8 | 73.6 | 70.6 |
| Iterative Attention (Ensemble) | 74.5 | 75.7 | 76.9 | 72.0 | 74.1 | **71.0** |
| **AoA Reader (Ensemble)** | - | - | **78.9** | **74.5** | **74.7** | 70.8 |
| **AoA Reader (Ensemble + Reranking)** | - | - | **80.3** | **75.6** | **77.0** | **74.1** |

Table 3: Results on the CNN news, CBTest NE and CN datasets. The best baseline results are depicted in italics, and the overall best results are in bold face.

Dimensions of embedding and hidden layer for each task are listed in Table 2. In re-ranking step, all language models are 8-gram, and trained by SRILM toolkit (Stolcke, 2002). The results are reported with the best model, which is selected by the performance of validation set. The ensemble model is made up of four best models, which are trained using different random seed. Implementation is done with Theano (Theano Development Team, 2016) and Keras (Chollet, 2015), and all models are trained on Tesla K40 GPU.

## 5.2 Overall Results

Our experiments are carried out on public datasets: CNN news datasets (Hermann et al., 2015) and CBTest NE/CN datasets (Hill et al., 2015). The statistics of these datasets are listed in Table 1, and the experimental results are given in Table 3.

As we can see that, our AoA Reader outperforms state-of-the-art systems by a large margin, where 2.3% and 2.0% absolute improvements over EpiReader in CBTest NE and CN test sets, which demonstrate the effectiveness of our model. Also by adding additional features in the re-ranking step, there is another significant boost 2.0% to

3.7% over AoA Reader in CBTest NE/CN test sets. We have also found that our single model could stay on par with the previous best ensemble system, and even we have an absolute improvement of 0.9% beyond the best ensemble model (Iterative Attention) in the CBTest NE validation set. When it comes to ensemble model, our AoA Reader also shows significant improvements over previous best ensemble models by a large margin and set up a new state-of-the-art system.

To investigate the effectiveness of employing *attention-over-attention* mechanism, we also compared our model to CAS Reader, which used pre-defined merging heuristics, such as *sum* or *avg* etc. Instead of using pre-defined merging heuristics, and letting the model explicitly learn the weights between individual attentions results in a significant boost in the performance, where 4.1% and 3.7% improvements can be made in CNN validation and test set against CAS Reader.

## 5.3 Effectiveness of Re-ranking Strategy

As we have seen that the re-ranking approach is effective in cloze-style reading comprehension task. To have a thorough investigation in the re-ranking

|              | CBTest NE |      | CBTest CN |      |
| ------------ | --------- | ---- | --------- | ---- |
|              | Valid     | Test | Valid     | Test |
| AoA Reader   | 77.8      | 72.0 | 72.2      | 69.4 |
| +Global LM   | 78.3      | 72.6 | 73.9      | 71.2 |
| +Local LM    | 79.4      | 73.8 | 74.7      | 71.7 |
| +Word-class LM | 79.6    | 74.0 | 75.7      | 73.1 |

Table 4: Detailed results of 5-best re-ranking on CBTest NE/CN datasets. Each row includes all of the features from previous rows. $LM_{global}$ denotes the global LM, $LM_{local}$ denotes the local LM, $LM_{wc}$ denotes the word-class LM.

step, we listed the detailed improvements while adding each feature mentioned in Section 4.

From the results in Table 4, we found that the NE and CN category both benefit a lot from the re-ranking features, but the proportions are quite different. Generally speaking, in NE category, the performance is mainly boosted by the $LM_{local}$ feature. However, on the contrary, the CN category benefit from $LM_{global}$ and $LM_{wc}$ rather than the $LM_{local}$.

|              | CBTest NE | CBTest CN |
| ------------ | --------- | --------- |
| NN           | 0.64      | 0.20      |
| Global LM    | 0.16      | 0.10      |
| Word-class LM | 0.04     | 0.39      |
| Local LM     | 0.16      | 0.31      |
| RATIO $\eta$ | 1.25      | 1.58      |

Table 5: Weight of each feature in N-best re-ranking step. *NN* denotes the feature (probability) produced by baseline neural network model.

Also, we listed the weights of each feature in Table 5. The $LM_{global}$ and $LM_{wc}$ are all trained by training set, which can be seen as *Global Feature*. However, the $LM_{local}$ is only trained within the respective document part of test sample, which can be seen as *Local Feature*.

$$\eta = \frac{LM_{global} + LM_{wc}}{LM_{local}} \quad (13)$$

We calculated the ratio between the global and local features and found that the NE category is much more dependent on local features than CN category. Because it is much more likely to meet a new named entity than a common noun in the test phase, so adding the local LM provides much more information than that of common

noun. However, on the contrary, answering common noun requires less local information, which can be learned in the training data relatively.

## 6 Quantitative Analysis

In this section, we will give a quantitative analysis to our AoA Reader. The following analyses are carried out on CBTest NE dataset. First, we investigate the relations between the length of the document and corresponding accuracy. The result is depicted in Figure 2.

As we can see that the AoA Reader shows consistent improvements over AS Reader on the different length of the document. Especially, when the length of document exceeds 700, the improvements become larger, indicating that the AoA Reader is more capable of handling long documents.

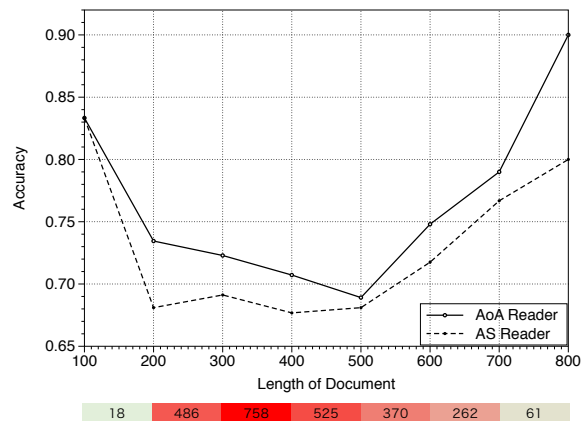

Figure 2: Test accuracy against the length of the document. The bar below the figure indicates the number of samples in each interval.

Furthermore, we also investigate if the model tends to choose a high-frequency candidate than a lower one, which is shown in Figure 3. Not surprisingly, we found that both models do a good job when the correct answer appear much frequent in the document than the other candidates. This is because that the correct answer that has the highest frequency among the candidates takes up over 40% of the test set (1071 out of 2500). But interestingly we have also found that, when the frequency rank of correct answer exceeds 7 (less frequent among candidates), these models also give a relatively high performance. Empirically, we think that these models tend to choose extreme cases in terms of candidate frequency (either too

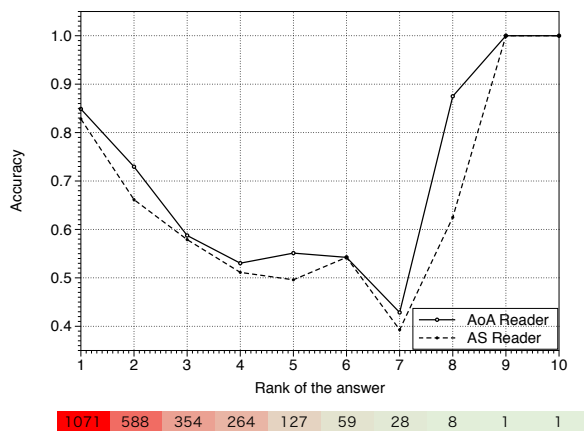

Figure 3: Test accuracy against the frequency rank of the answer. The bar below the figure indicates the number of samples in each rank.

high or too low). One possible reason is that the model is hard to choose a candidate that has a neutral frequency as the correct answer, because of its ambiguity (neutral choices are hard to made).

## 7 Related Work

Cloze-style reading comprehension tasks have been widely investigated in recent studies. We will take a brief revisit to the related works.

Hermann et al. (2015) have proposed a method for obtaining large quantities of $\langle \mathcal{D}, \mathcal{Q}, \mathcal{A} \rangle$ triples through news articles and its summary. Along with the release of cloze-style reading comprehension dataset, they also proposed an attention-based neural network to handle this task. Experimental results showed that the proposed neural network is effective than traditional baselines.

Hill et al. (2015) released another dataset, which stems from the children's books. Different from Hermann et al. (2015)'s work, the document and query are all generated from the raw story without any summary, which is much more general than previous work. To handle the reading comprehension task, they proposed a window-based memory network, and self-supervision heuristics is also applied to learn hard-attention.

Unlike previous works, that using blended representations of document and query to estimate the answer, Kadlec et al. (2016) proposed a simple model that directly pick the answer from the document, which is motivated by the Pointer Network (Vinyals et al., 2015). A restriction of this model is that the answer should be a single word

and appear in the document. Results on various public datasets showed that the proposed model is effective than previous works.

Liu et al. (2016) proposed to exploit reading comprehension models to other tasks. They first applied the reading comprehension model into Chinese zero pronoun resolution task with automatically generated large-scale pseudo training data. The experimental results on OntoNotes 5.0 data showed that their method significantly outperforms various state-of-the-art systems.

Our work is primarily inspired by Cui et al. (2016) and Kadlec et al. (2016) , where the latter model is widely applied to many follow-up works (Sordoni et al., 2016; Trischler et al., 2016; Cui et al., 2016). Unlike the CAS Reader (Cui et al., 2016), we do not assume any heuristics to our model, such as using merge functions: $sum$, $avg$ etc. We used a mechanism called "attention-over-attention" to explicitly calculate the weights between different individual document-level attentions, and get the final attention by computing the weighted sum of them. Also, we find that our model is typically general and simple than the recently proposed model, and brings significant improvements over these cutting edge systems.

## 8 Conclusion

We present a novel neural architecture, called attention-over-attention reader, to tackle the cloze-style reading comprehension task. The proposed AoA Reader aims to compute the attentions not only for the document but also the query side, which will benefit from the mutual information. Then a weighted sum of attention is carried out to get an attended attention over the document for the final predictions. Among several public datasets, our model could give consistent and significant improvements over various state-of-the-art systems by a large margin.

The future work will be carried out in the following aspects. We believe that our model is general and may apply to other tasks as well, so firstly we are going to fully investigate the usage of this architecture in other tasks. Also, we are interested to see that if the machine really "comprehend" our language by utilizing neural networks approaches, but not only serve as a "document-level" language model. In this context, we are planning to investigate the problems that need comprehensive reasoning over several sentences.

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
