# Peer review of "Attention-over-Attention Neural Networks for Reading Comprehension"

_ACL 2017 — decision unknown_

[Official Review · Reviewer 1 · rating 5 · confidence 4]
soundness 5 · originality 3 · clarity 5 · substance 4 · appropriateness 5 · presentation format Oral Presentation

- Strengths:

-- A well-motivated approach, with a clear description and solid results.

- Weaknesses:

-- Nothing substantial other than the comments below. 

- General Discussion:

The paper describes a new method called attention-over-attention for reading
comprehension. First layers of the network compute a vector for each query word
and document word, resulting in a |Q|xK matrix for the query and a |D|xK for
the document. Since the answer is a document word, an attention mechanism is
used for assigning weights to each word, depending on their interaction with
query words. In this work, the authors deepen a traditional attention mechanism
by computing a weight for each query word through a separate attention and then
using that to weight the main attention over document words. Evaluation is
properly conducted on benchmark datasets, and various insights are presented
through an analysis of the results as well as a comparison to prior work. I
think this is a solid piece of work on an important problem, and the method is
well-motivated and clearly described, so that researchers can easily reproduce
results and apply the same techniques to other similar tasks.

- Other remarks:

-- p4, Equation 12: I am assuming i is iterating over training set and p(w) is
referring to P(w|D,Q) in the previous equation? Please clarify to avoid
confusion.

-- I am wondering whether you explored/discussed initializing word embeddings
with existing vectors such as Google News or Glove? Is there a reason to
believe the general-purpose word semantics would not be useful in this task?

-- p6 L589-592: It is not clear what the authors are referring to when they say
'letting the model explicitly learn weights between individual attentions'? Is
this referring to their own architecture, more specifically the GRU output
indirectly affecting how much attention will be applied to each query and
document word? Clarifying that would be useful. Also, I think the improvement
on validation is not 4.1, rather 4.0 (72.2-68.2).

-- p7 Table 5: why do you think the weight for local LM is relatively higher
for the CN task while the benefit of adding it is less? Since you included the
table, I think it'll be nice to provide some insights to the reader.

-- I would have liked to see the software released as part of this submission.

-- Typo p2 L162 right column: "is not that effective than expected" --> "is not
as effective as expected"?

-- Typo p7 L689 right column: "appear much frequent" --> "appears more
frequently"?

-- Typo p8 L719-721 left column: "the model is hard to" --> "it is hard for the
model to"? & "hard to made" --> "hard to make"?